# Allopregnanolone Enhances GABAergic Inhibition in Spinal Motor Networks

**DOI:** 10.3390/ijms21197399

**Published:** 2020-10-07

**Authors:** Berthold Drexler, Julia Grenz, Christian Grasshoff, Bernd Antkowiak

**Affiliations:** Experimental Anesthesiology Section, Department of Anesthesiology and Intensive Care Medicine, Eberhard-Karls-University, Waldhörnlestrasse 22, 72072 Tübingen, Germany; berthold.drexler@uni-tuebingen.de (B.D.); julia.grenz@med.uni-heidelberg.de (J.G.); christian.grasshoff@uni-tuebingen.de (C.G.)

**Keywords:** allopregnanolone, spinal networks, electrophysiology, organotypic cultures, neuro-muscular junction, propofol, mechanisms of anesthesia

## Abstract

The neurosteroid allopregnanolone (ALLO) causes unconsciousness by allosteric modulation of γ-aminobutyric acid type A (GABA_A_) receptors, but its actions on the spinal motor networks are unknown. We are therefore testing the hypothesis that ALLO attenuates the action potential firing of spinal interneurons and motoneurons predominantly via enhancing tonic, but not synaptic GABAergic inhibition. We used video microscopy to assess motoneuron-evoked muscle activity in organotypic slice cultures prepared from the spinal cord and muscle tissue. Furthermore, we monitored GABA_A_ receptor-mediated currents by performing whole-cell voltage-clamp recordings. We found that ALLO (100 nM) reduced the action potential firing of spinal interneurons by 27% and that of α-motoneurons by 33%. The inhibitory effects of the combination of propofol (1 µM) and ALLO on motoneuron-induced muscle contractions were additive. Moreover, ALLO evoked a tonic, GABA_A_ receptor-mediated current (amplitude: 41 pA), without increasing phasic GABAergic transmission. Since we previously showed that at a clinically relevant concentration of 1 µM propofol enhanced phasic, but not tonic GABAergic inhibition, we conclude that ALLO and propofol target distinct subpopulations of GABA_A_ receptors. These findings provide first evidence that the combined application of ALLO and propofol may help to reduce intraoperative movements and undesired side effects that are frequently observed under total intravenous anesthesia.

## 1. Introduction

Considerable research efforts are focusing on neuroactive steroids as adjuncts or even substitutes for commonly used intravenous anesthetics [1,2], which have a long list of adverse side effects [3]. The endogenous neurosteroid allopregnanolone (ALLO) is part of the body’s own neuroprotective response, triggered by pregnancy, injury, and stress. It is synthesized inside and outside the central nervous system, is able to cross the blood-brain-barrier [4], and has profound hypnotic properties [5,6]. As a component of the body’s own neurosteroid synthesis, it is thought to produce fewer side effects than exogenous substances. ALLO acts as a positive allosteric modulator on γ-aminobutyric acid type A receptors (GABA_A_-R) [7,8] and, when applied in the high concentration range, directly opens GABA_A_-Rs. These ligand-gated ion channels assemble from five protein subunits and are permeable to chloride and bicarbonate ions. In mammals, nineteen different subunits of the GABA_A_-R are expressed (α1–6, β1–3, γ1–3, δ, ε π, ς1–3), giving rise to a large number of different receptor subtypes [9]. Subtypes of the GABA_A_-R residing at synapses are involved in rapid phasic inhibition, whereas extrasynaptic receptors, which are sensitive to very low concentrations of ambient GABA, are thought to produce a tonic form of inhibition [10]. A major population of extrasynaptic GABA_A_-Rs harbors δ-subunits [11]. Interestingly, δ-subunits containing receptors are highly sensitive to neuroactive steroids and are expressed not only in the brain but also in the spinal cord [12]. Several lines of evidence suggest that anesthetic-induced muscle relaxation and ablation of movements in response to a noxious stimulus is mediated primarily by spinal neurons [13,14]. Hitherto, we know very little about the actions of neurosteroids on spinal motor circuits. Thus, our study tests the hypothesis that ALLO attenuates action potential firing of spinal interneurons and motoneurons predominantly via enhancing tonic but not synaptic GABAergic inhibition. Furthermore, we address the question whether ALLO and propofol, the most frequently used intravenous anesthetics [15], affect spinal motoneurons in a synergistic manner as previously reported for synaptically-mediated GABAergic currents in cultured neurons from the neocortex [16]. In the present work, we utilize organotypic co-cultures of the spinal cord and muscle tissue to explore the effects of ALLO on spinal motor networks. This preparation’s morphological features and functional properties were described in greater detail in the landmark papers published by Spenger et al. as well as by Streit et al. These authors provided evidence that spinal interneurons and motoneurons, neuromuscular junctions, and stratified muscle fibers ex vivo largely resemble their counterparts in vivo [17]. These researchers identified motoneurons by intracellular filling and retrograde axonal staining, using horseradish peroxidase crystals applied to the muscle tissue. Furthermore, neuromuscular junctions were visualized as acetylcholinesterase-positive patches on muscle fibers, and functional evidence for the formation of spinal reflex arcs was provided [18]. Moreover, a study on multiple developmentally regulated markers showed close similarities to those observed in vivo, suggesting that spinal organotypic cultures provide a useful ex vivo model to study several aspects of neurogenesis and muscle innervation [19]. Our group makes use of this preparation to study the effects of anesthetic agents (e.g., propofol, etomidate, thiopental, and sevoflurane) [20], muscle relaxants [21], and neurotoxins [22,23,24] to further elucidate the drugs’ mechanisms of action and to develop cell-based ex vivo assays in order to reduce the need for animal testing [23,24].

The major conclusions drawn from our current study using this type of model system are that, at nanomolar concentrations, ALLO significantly decreased action potential firing of spinal interneurons and motoneuron-induced muscle contractions predominantly by enhancing tonic but not synaptic GABA_A_-R-mediated inhibition. Furthermore, the effects of ALLO and propofol on muscle activity appeared to be additive, suggesting that these agents may act via different subtypes of GABA_A_-Rs.

## 2. Results

In the present study, we quantified GABA_A_ receptor-mediated currents by means of whole-cell voltage-clamp recordings, carried out in 32 organotypic slice cultures derived from the spinal cord of embryonic C57bl6 mice. Furthermore, we monitored action potential activity of spinal neurons in 483 tissue slices, using extracellular glass-microelectrodes. We also made video recordings of muscle contractions in 57 co-cultures. To quantify the effects of solvents (sham-application) we used another 133 slices.

### 2.1. Actions of Allopregnanolone on Tonic GABAergic Inhibition

We voltage-clamped cells at −70 mV, exposed them to ALLO and subsequently treated them with the specific GABA_A_ antagonist bicuculline (20 µM) to probe for actions of ALLO on tonic GABAergic currents in spinal cord neurons. Representative results from a typical recording are displayed in Figure 1, showing the shift in the baseline current caused by the specific GABA_A_ antagonist bicuculline (Figure 1A) and the corresponding all-point-histogram (Figure 1B). Figure 1C summarizes the changes in tonic currents caused by three different concentrations of ALLO. Amplitudes of bicuculline-sensitive currents were 40.8 ± 9.8 pA (100 nM) and 49.3 ± 8.2 pA (250 nM) and statistically different from the current in the presence of 50 nM ALLO (9.7 ± 5.3 pA, n.s.).

### 2.2. Actions of Allopregnanolone on Phasic GABAergic Inhibition

Next, we explored ALLO’s effects on GABA_A_ receptor-dependent synaptic transmission by evaluating the changes in the frequency and the kinetic properties of inhibitory postsynaptic currents (IPSCs) derived from whole-cell voltage-clamp recordings. Representative current traces are shown in Figure 2A. At a concentration of 100 nM, ALLO caused a small, yet statistically significant, increase in the IPSC decay time. We detected this effect in both the weighted time constant of biphasically fitted synaptic events (Figure 2B: control: 25.2 ms, ALLO: 31.1 ms, *n* = 11, *t*-test: *p* = 0.0035) and in the width of events at half-maximal amplitudes (Figure 2C: control: 13.1 ms, ALLO: 14.7 ms, *n* = 11, *t*-test: *p* = 0.024). Simultaneously, ALLO (100 nM) did slightly reduce the amplitudes of IPSCs (Figure 2D) and decreased their frequency (Figure 2E: control: 9.1 Hz, ALLO: 6.1 Hz, *n* = 11, *t*-test: *p* = 0.0005). The charge transferred during average IPSCs and the total synaptic current (calculated by multiplying the charge per average IPSC by the frequency of occurrence) neither differed in the absence nor in the presence of ALLO. The effects observed after exposing the cells to 250 nM ALLO were almost identical to those observed at a concentration of 100 nM (Figure A1).

### 2.3. Effect of Allopregnanolone on Action Potential Firing of Spinal Neurons

In another series of experiments, we used extracellular microelectrodes to study the effects of ALLO on the spontaneous discharge rate of interneurons in the ventral part of the spinal tissue in 257 cultured slices. The cells we used generated fast sodium-dependent action potentials at a frequency of 9.6 ± 1.3 Hz prior to ALLO treatment. The effects of different concentrations of ALLO, ranging from 50 nM to 2 µM, are provided in Figure 3 as normalized activities. We obtained normalized activities by dividing the firing rate in the presence of ALLO by the one obtained prior to the treatment. Thus, a normalized activity of 1 indicates no effect at all and a normalized activity of 0.5 corresponds to a 50% depression.

ALLO significantly inhibited action potential firing, starting from 100 nM to 0.73 (0.47, 0.90) of control activity to 0.54 (0.43, 0.62) at 2 µM (*n* = 35–48 per concentration). Normalized activities calculated from sham-applications were very close to 1, indicating that the solvent was without effect. It is worth pointing out that the concentration-dependent switch towards significant drug actions occurred from 50 nM to 100 nM, mirroring the concentration-dependent changes in tonic GABA_A_ receptor-mediated currents induced by ALLO (see Figure 1C).

### 2.4. Actions of Allopregnanolone on Muscle Activity in Spinal Nerve and Muscle Co-Cultures

In co-cultures of the spinal cord and muscle tissue, α-motoneurons and striated muscle fibers are synaptically connected and action potentials in α-motoneurons translate into contractions of muscle fibers. We determined the frequency of muscle contractions by the use of videomicroscopy to assess whether ALLO and propofol reduce the activity of α-motoneurons. The results displayed in Figure 4 provide evidence that in our preparation, the occurrence of muscle contractions correlates with the action potential activity of spinal neurons. Under drug-free conditions, simultaneous recordings of action potentials in the ventral horn of the spinal cord and muscle contractions showed a rhythmic activity and a stable phase relationship between these signals (Figure 4A). A video showing simultaneous action potential firing and muscle contractions in organotypic co-cultures of the spinal cord and muscle tissue can be found in the Appendix A accompanying this paper. The application of pancuronium, a non-depolarizing muscle relaxant that acts as a specific competitive antagonist at nicotinic acetylcholine receptors, abolished muscle contractions, leaving the activity pattern of spinal neurons largely unaffected (Figure 4B). This observation suggests that nicotinic acetylcholine receptors play a minor role in controlling the firing of spinal neurons [25] but a major role in neuromuscular synapses. Moreover, the effect of pancuronium was fully reversible (Figure 4C). The application of tetrodotoxin, a specific blocker of voltage-dependent sodium channels, suppressed fast action potentials of spinal neurons and muscle contractions (Figure 4D). After the drug’s removal, both the activity of spinal neurons and the muscle contractions reappeared (Figure 4E). In all experiments aiming to quantify the muscle relaxant properties of propofol and ALLO, we applied pancuronium (1 µM) at the end of our recordings to prove that muscle activity was caused by α-motoneurons.

The effects of ALLO and propofol and a combination of these two agents on the frequency of muscle contractions are summarized in Figure 5. At a concentration of 1 µM, propofol decreased muscle contractions by 23% (control activity: 1.2 (0.8, 1.3) Hz; 1 µM propofol: 0.92 (0.55, 1.04) Hz), which is very close to the anesthetic’s previously reported effect on spinal neurons in the ventral part of cultured spinal slices (inhibition of the action potential firing of 25% [26]). ALLO (100 nM) on its own reduced the muscle contraction frequency to 0.81 (0.70, 0.95) Hz, corresponding to an inhibition of 32.5%. The combined application of ALLO (100 nM) and propofol (1 µM) further reduced the median muscle contraction frequency to 0.56 (0.32, 0.78) Hz. Further, we observed that the additional as well as the sole presence of pancuronium led to an almost complete relaxation of muscular fibers.

The pharmacological interactions between ALLO and propofol are summarized in Figure 6. For the sake of clarity, only median activities and no error bars are displayed. Furthermore, we included the data obtained by exposing co-cultures to a rather high propofol concentration of 5 µM. Our first finding is that the combined application of 100 nM ALLO and 1 µM propofol is similarly effective in reducing the activity of α-motoneurons (inhibition: 51.5%) compared with 5 µM propofol (inhibition: 52%). Our second observation is that ALLO’s capacity to reduce muscle activity is fully maintained by 1 µM propofol but largely reduced by 5 µM propofol.

## 3. Discussion

### 3.1. Allopregnanolone Induces a Tonic GABA_A_-R-Mediated Current

To our best knowledge, the present study shows, for the first time, that ALLO induces a tonic, GABA_A_-R-mediated current in spinal interneurons of the ventral horn, which is accompanied by a decrease in motor output. This tonic current may involve the activation of extrasynaptic GABA_A_ receptors. A subpopulation of extrasynaptic GABA_A_ receptors contains δ-subunits. In mouse dentate gyrus granule cells, ALLO, when applied at nanomolar concentrations, amplified GABA induced currents [27]. This potentiation was substantially lessened in neurons derived from δ-subunit-knockout animals, indicating that δ-GABA_A_-Rs are highly sensitive to ALLO. In the spinal cord, δ-subunits are present in the dorsal horn and are involved in nociception [12], but their expression in motor circuits is uncertain. The δ-subunit and the α6-subunit are frequently partnered in extrasynaptic GABA_A_-Rs [9] and are highly sensitive to furosemide [28]. In the turtle spinal cord, furosemide-sensitive tonic currents were detected in the majority of tested motoneurons [29] but not in ventral horn interneurons [30]. Taken together, these findings do not support the idea that the bicuculline-sensitive tonic current evoked by ALLO in the present study involves δ-containing GABA_A_-Rs. Another well-known candidate for mediating tonic currents in the spinal cord are GABA_A_-Rs harboring α5-subunits [31]. Castro et al. detected a tonic current in ventral horn interneurons that was blocked by the specific α5-antagonist L−655,708 [30]. The existence of this specific GABA_A_-R subtype in these neurons was further confirmed by RT-PCR and immunohistochemistry. Tonic GABAergic currents that were sensitive to L−655,708 were identified in motoneurons as well [32]. The conclusion that α5-GABA_A_-Rs are involved in controlling spinal motor networks also in vivo was supported by the finding that L−655,708 modulates the Hoffmann reflex in healthy and diabetic rats [33].

The mechanism by which ALLO enhances tonic GABAergic currents on the molecular level remains to be elucidated. ALLO may act by positive allosteric modulation of agonist-induced receptor activation caused by ambient GABA. In this case, extrasynaptic GABA may arise from synaptically released GABA, and from GABA delivered by glial cells [34]. However, ALLO-induced tonic currents may also be caused by direct activation of GABA_A_-Rs, a mechanism that has been proposed for the bicuculline-sensitive current induced by the intravenous anesthetic thiopental [35].

### 3.2. Actions of Allopregnanolone on Muscle Contractions

In the present study, ALLO significantly decreased pancuronium-sensitive muscle contraction. Several lines of evidence suggest that these contractions were induced by α-motoneurons. First, retrograde axonal staining, using horseradish peroxidase crystals topically applied to the muscle tissue exclusively labeled putative motoneurons characterized by large, acetylcholinesterase-positive neurons, located at the ventral border of cultured spinal tissue slices [17]. These observations suggest that muscle fibers were only innervated by motoneurons. Second, simultaneous recordings of action potential firing of ventral horn neurons and muscle contractions revealed that pancuronium, a nicotinic antagonist and muscle relaxant drug, only inhibited muscle contractions but not the discharge rate of spinal neurons (see Figure 4), indicating a mechanism of action downstream spinal motoneurons. Third, low-intensity electrical stimulation of mouse diaphragm muscles revealed that pancuronium, at a concentration of 1 µM, depressed synaptically-mediated muscle contractions but was ineffective to block contractions resulting from direct electrical stimulation of muscle fibers [36]. Taken together, these findings strongly implicate that pancuronium-sensitive muscle contractions as reported here were caused by α-motoneurons. We found that ALLO significantly decreased neurogenic muscle contractions. This action was fully preserved in spinal slices exposed to 1 µM propofol, a concentration causing unconsciousness in humans and rodents [37]. On a molecular level, ALLO acted by producing a tonic GABA_A_ receptor-mediated current with minor effects on synaptically-mediated GABAergic inhibition. In a previous study, we showed that in contrast to ALLO, propofol administered at a clinically relevant concentration of 1 µM failed to induce a tonic GABAergic current in spinal neurons, but, again in contrast to ALLO, significantly enhanced synaptically-mediated GABAergic inhibition. Together, these observations prompt the hypothesis that ALLO and propofol target different sub-populations of GABA_A_ receptors. Dissimilar molecular targets of propofol and ALLO may also explain their almost perfect additive interaction on α-motoneurons (100 nM ALLO: 32.5% median inhibition; 1 µM propofol: 23.3% inhibition; estimated additive effect: 55.8%; experimentally induced additive effect: 51.5%). This changed, however, when we increased propofol’s concentration to 5 µM. This very high concentration which causes immobility in humans [38], evoked a tonic bicuculline-sensitive current, similar to ALLO. Interestingly, the effect of ALLO on α-motoneurons was substantially smaller in slices exposed to 5 µM propofol compared to slices treated with only 1 µM propofol. The underlying mechanisms are subject to speculation. It seems unlikely that propofol displaces ALLO from its binding site on GABA_A_ receptors or vice versa, since the neurosteroid binding site is different from the binding sites of propofol [8]. Furthermore, there is evidence in the literature that both ALLO and intravenous anesthetics can be present at their binding sites on GABA_A_ receptors at the same time [39]. Another explanation is that the activation of presumably extrasynaptic GABA_A_ receptors causing the tonic current was almost in the saturating range. This hypothesis is indeed supported by the findings that only an increase in the concentration of ALLO from 50 nM to 100 nM, not, however, from 100 to 250 nM substantially enhanced this current. Furthermore, the amplitudes of tonic currents induced by 5 µM propofol and ALLO were quite similar (around 45 pA).

### 3.3. Effects of Allopregnanolone on Synaptic GABA_A_ Receptors Are of Minor Importance

Besides inducing a tonic GABAergic current, ALLO also modified GABAergic IPSCs by increasing their decay time by about 25% (Figure 2) and by decreasing the frequency of synaptic events without causing statistically significant changes in their amplitudes. A prolonged action of ALLO on GABAergic IPSCs was reported in dorsal horn neurons but at much higher concentrations ranging from 1 µM to 10 µM [40]. This raised the question whether it is likely that the effects of ALLO on the kinetics of IPSCs seen in this study substantially contribute to the drug’s inhibitory effect on action potential firing. This issue can be addressed by comparing the total amount of charge carried by tonic and phasic GABAergic currents. In our study, the charge transferred on average in the course of a single IPSC was about 0.7 pC (Figure 2F), and the frequency of IPSCs was approximately 10 Hz (Figure 2E). This corresponds to a tonic current of 7 pA flowing across synaptic GABA_A_ receptors. Assuming that ALLO increased the charge transferred per IPSC by 25% without changing the amplitude and frequency, the respective current induced by the neurosteroid is about 1.75 pA. However, at the same time, ALLO (100 nM) induced a tonic GABAergic current of about 40 pA (Figure 1C). This rough estimation, not taking into account that ALLO reduced the frequency of IPSCs (Figure 2E), suggests that the effects of ALLO on synaptic GABAergic transmission did not significantly contribute to the drug’s inhibitory effect on the excitability of spinal neurons.

### 3.4. Comparison of Spinal and Cortical Actions of Allopregnanolone

A previous study explored the effects of ALLO in cultured neocortical slices [16]. At a concentration of 100 nM, ALLO substantially prolonged the decay of GABAergic IPSCs by about 70%, compared to only 25% in spinal neurons (Figure 2B,C). However, the decrease in the action potential firing caused by ALLO (100 nM) was almost identical in both the spinal and the neocortical neurons. This suggests a less prominent enhancement of tonic GABAergic currents in cortical neurons than observed in spinal neurons. These results provide further support for non-uniform effects caused by neurosteroids in different regions of the CNS. Region-specific effects can be explained by network-specific expression of certain GABA_A_ receptor subunits and by ALLO’s preference of certain GABA_A_ receptor subtypes over others [3]. Indeed, studies on expressed receptors showed that neurosteroids mainly interact with GABA_A_ receptors containing α1 and α3 subunits, whereas their actions on GABA_A_ receptors containing α2, α4, α5, and α6 subunits are rather weak [41]. Furthermore, phosphorylation of synaptic GABA_A_ receptors can have dramatic effects on neurosteroid binding [42].

### 3.5. Limitations of the Present Study

It is important to mention the limitations of the present study. We have observed a good correlation between the concentration-dependent enhancement of tonic GABAergic currents (Figure 1C) and the depression of spontaneous action potential firing of ventral horn interneurons (Figure 3). However, experimental conditions for recording action potentials and GABAergic current were different. In the latter case, we blocked glutamatergic and glycinergic neurotransmission. This intervention interrupted synaptic interactions in the network of spinal interneurons and reduced glutamatergic excitation of GABA-releasing neurons, which potentially impacts the balance between phasic and tonic GABAergic inhibition. Thus, technically it seems possible that ALLO might potentiate GABAergic synaptic transmission in the case of preserved glutamatergic neurotransmission. On the other hand, the large size of the extracellular space and continuous perfusion with artificial cerebrospinal fluid is probably associated with extracellular GABA-concentrations much lower than those occurring in vivo. Another limitation is that the present study assessed drug-induced changes in spontaneous action potential firing but not stimulus induced neuronal activity and muscle contractions as these different types of activity might involve different neuronal networks.

### 3.6. Implications for Further Studies

As outlined above, there is now evidence that bicuculline-sensitive tonic currents are modulating the activity of spinal ventral interneurons and motoneurons. Specifically, pharmacological studies are suggesting a role of α5- and α6-subunit containing GABA_A_-R in producing these inhibitory currents. However, the GABA_A_-R subtypes targeted by neuroactive steroids in spinal motor networks remain to be identified. For addressing this issue in future studies, actions of neuroactive steroids should be characterized in tissue slices derived from α5- and δ-subunit knockout mice. In a following step, hypotheses arising from these in vitro experiments can be tested in vivo by investigating ALLO’s effects on the Hoffmann reflex, which involves monosynaptic stimulation of α-motoneurons.

## 4. Materials and Methods

### 4.1. Preparation of Nerve and Muscle Co-Cultures

All procedures were approved by the local Animal Care Committee (Eberhard-Karls-University, Tübingen, Germany, 1 March 2017) and were performed in accordance with the German Animal Welfare Act (TierSchG). We made every endeavor to minimize both the suffering and the number of animals we used for this study. We sacrificed pregnant C57/BL6J mice (E 13–15) under deep anesthesia to obtain 300 µm embryonic tissue slices from the spinal cord and surrounding muscles as previously reported [43]. We fixed these slices on a glass coverslip, using a mixture of chicken plasma and thrombin, and transferred them into a plastic tube supplemented with 0.75 mL of a nutrition medium enhanced with l-glutamine and glucose. The medium was composed of 25 vol-% horse serum, 25 vol-% Hank’s balanced salt solution and 50 vol-% Basal Medium Eagle. Furthermore, we added a neuronal growth factor to obtain a final concentration of 10 nM. After three days in culture, we added antimitotics (10 µM 5-fluoro-2-deoxyuridine, 10 µM cytosine-b-arabinofuranoside, 10 µM u ridine) to reduce the growth of glial cells. We applied the roller tube approach to culture the tissue [44]. In approximately two thirds of the cultures spontaneous muscle contractions developed within the first week ex vivo, indicating a de novo formation of neuromuscular junctions. We recorded muscle contractions in cultures after 23 days ex vivo. By then these cultures had passed major developmental steps in maturation [17,18,19,45].

### 4.2. Preparation and Application of Test Solutions

We dissolved ALLO in DMSO to a 1 mM stock solution. For the stock solution with pancuronium bromide, we used distilled water. Prior to the experiment, we diluted ALLO, pancuronium, and propofol in ACSF to reach our target concentration. The drug containing ACSF was applied via bath perfusion at a flow rate of 1 mL min^−1^. When switching from ACSF to drug-containing ACSF, the medium in the experimental chamber was replaced by at least 95% within 2 min. Twelve minutes after changing to the drug-containing perfusate, we started our recordings for the duration of the drug treatment. This time interval has been proven to be sufficient to adjust steady-state conditions, as diffusion times in slice cultures are considerably shorter compared to acute slice preparations [46,47]. We obtained almost all chemicals for our study from Sigma-Aldrich, (Taufkirchen, Germany). The horse serum came from Invitrogen, (Karlsruhe, Germany) and the propofol from Fresenius Kabi (Bad Homburg, Germany).

### 4.3. Recording and Analysis of Electrophysiological Data

For our extracellular network recordings, we used a recording chamber mounted on an inverted microscope (Carl Zeiss Microimaging, Göttingen, Germany). We perfused slices with ACSF consisting of (in mM) NaCl 120, KCl 3.3, NaH_2_PO_4_ 1.13, NaHCO_3_ 26, CaCl_2_ 1.8, MgCl_2_ 1.0, glucose 11, with 95% oxygen and 5% carbon dioxide.

The recording chamber consisted of a metal frame with a glass bottom and had a volume of 1.5 mL. We conducted all experiments at 34 °C. We visually identified the ventral horn of the spinal cord and inserted ACSF-filled glass electrodes with a resistance of about 3 to 5 MΩ into the tissue until we detected extracellular spike activity exceeding 100 µV in amplitude. We acquired all data on a personal computer, using the digidata 1200 AD/DA interface and the Axoscope 9.0.1 software (Molecular Devices, San Jose, CA, USA). For the filtering of extracellularly recorded signals and the counting of action potentials, we used self-written programs in Matlab 7.6 (The Mathworks, Natick, MA, USA). To compute the average action potential firing rate, we utilized an automated event detection algorithm with a threshold set well above the baseline noise.

We performed intracellular recordings with a Multiclamp 700 A (Molecular Devices, San Jose, CA, USA) amplifier at room temperature. The extracellular medium consisted of ACSF as outlined above in the additional presence of 40 µM dl-2-amino-5-phosphonopentanoic-acid, 15 µM 6-cyano-7-nitro-quinoxaline-2,3-dione, (Tocris, Minneapolis, MN, USA) and 1 µM strychnine to block glutamatergic and glycinergic currents. We identified interneurons in the ventral horn of the spinal cord part of the slice culture according to their morphology on a monitor using infrared illumination and a 40× water immersion objective. We used patch pipettes made from borosilicate glass (World Precision Instruments, Sarasota, FL, USA) with a p-2000 laser puller (Sutter Instruments, Novato, CA, USA) and filled them with a recording solution containing (in mM): CsCl 121, CsOH 24, MgCl_2_ 1, EGTA 5, HEPES 10, ATP 4 at pH 7.2 had a resistance of approximately 1.5–3.5 MΩ. We sampled whole-cell voltage clamp recordings from neurons held at –70 mV at 10 kHz, using the Digidata 1440 A interface and the Clampex 10.4 software (Molecular Devices, San Jose, CA, USA). To analyze intracellular data, we used self-written routines in Matlab 7.6. For the analysis of tonic currents, we applied the method proposed by Glykys and Mody [48].

### 4.4. Recording and Analysis of Muscle Activity

After optimizing contrast and brightness, we monitored muscle movements at an optic magnification of 320 before and during drug-exposure with a video camera (DMK21AU04, The Imaging Source, Bremen, Germany) and stored in an avi-format. We made recordings at a sample frequency of 30 Hz for each condition (control, drug, wash). We quantified the muscle activity offline, using a Hewlett Packard Z800 video-workstation (Picturetools, Hamburg, Germany). First, we defined the regions of interest (ROI) by visual inspection, making sure that these ROI contained high-contrast borders of muscle fibers which were dislocated in the course of the muscle contractions. Due to the muscle movements, the brightness of pixels located within the ROI changed over time. These changes well correlated with the intensity and frequency of the muscle contractions. To analyze the data we used software written in Matlab 7.6. Further details can be found in reference [21,23,24]. For reducing data variability we applied a quality-controlled algorithm that was based on multiple criteria. The first criterion was the minimum frequency of muscle contractions under drug-free conditions. We used a time window of 360 s for performing video recordings and for estimating the median frequency of muscle contractions. Under drug-free conditions, the median frequency was roughly 1 Hz and dropped to about 0.5 Hz in the presence of ALLO and propofol (see Figure 5 and Figure 6). Under both conditions, more than 100 events were counted during a single video recording. When recordings were repeated without changing drug conditions, the resulting medians showed little variability. However, in a subset of preparations, the frequency of spontaneous muscle contractions was, under drug free conditions, only around 0.01 Hz, corresponding to less than five contractions sampled during a single recording session. When repeating recordings in these preparations without changing drug conditions, the variability of the medians was large, indicating that the number of sampled events was too small for making reliable estimations of muscle activity. Accordingly, only preparations displaying a median frequency of muscle contractions of at least 0.1 Hz were used for further analysis. The second criterion was that shifting the threshold for detecting muscle contractions up and down did not affect the number of sampled events. Initially, the threshold for detecting the transitions of single muscle fibers from the relaxed to the contracted state was set to 50% of the associated change in brightness. In the majority of recordings, shifting the threshold upwards and downwards by 20% of the signal amplitude did not affect the number of counted muscle contractions. However, in a minority of recordings, this condition was not met. Since recordings were taken from single fibers, we conclude that the video signal was contaminated with noise and respective recordings were excluded from further analysis. The third criterion was related to the occurrence of tonic muscle contractions. In some cases, long-lasting tonic muscle contractions were observed under drug-free conditions. Because in these samples spinal neurons seemed to suffer from over-excitation and the frequency of muscle contractions did not represent the degree of muscle activity, these preparations were excluded from further analysis. The fourth criterion considered the sensitivity of muscle contraction to the non-depolarizing muscle relaxant pancuronium. In the present work, the frequency of muscle contractions was interpreted as a correlate of action potential firing of spinal motoneurons. This assumption is based on the observation that acetylcholinesterase-positive neurons, located at the ventral border of spinal slices and displaying typical morphological properties of motoneurons were labelled by retrograde axonal staining using horseradish peroxidase crystals selectively applied to the muscle tissue [17]. In the present study, the non-depolarizing muscle relaxant pancuronium was applied at the end of every single experiment in order to verify that muscle contractions were initiated by cholinergic neuromuscular transmission. Pancuronium, when applied at a concentration of 1 µM suppressed synaptically driven contraction in isolated mouse diaphragms but not contractions evoked by direct electrical stimulation of muscle fibres [36]. We found that in about 20% of tissue cultures, pancuronium failed to depress the frequency of muscle contractions, suggesting that these contractions were intrinsic in origin and did not involve transmission at the neuromuscular endplate. These recordings were removed from further analysis.

### 4.5. Statistical Analysis

We used the statistic toolbox of Matlab 7.6. for statistical testing. We conducted the Lilliefors test to test the distribution of the data. Data distributed in a normal range are given as mean ± SEM; for statistical comparison, we used the Student’s *t*-test. If normal distribution of data was rejected, data are given as boxplots (line: median, box: lower quartile = 25th percentile and upper quartile = 75th percentile; whisker: 1.5*iqr); for statistical testing we used the Mann-Whitney-U-test with Bonferroni correction; *p*-values smaller than 0.05 were considered as significant.

## Figures and Tables

**Figure 1 ijms-21-07399-f001:**
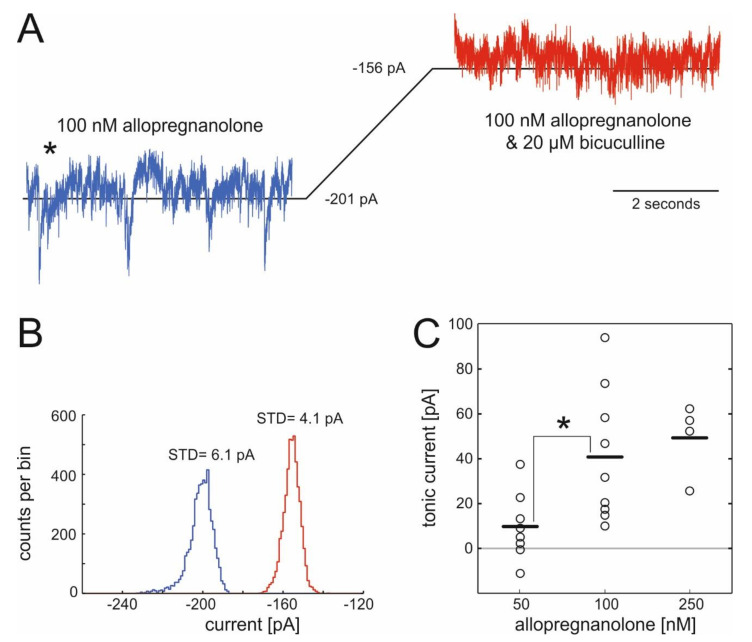
Actions of allopregnanolone on tonic currents in spinal interneurons of the ventral horn. (**A**) Example of a whole-cell voltage-clamp recording made from a spinal interneuron in the ventral horn at a holding potential of −70 mV in the presence of AP-V (dl-2-amino-5-phosphonopentanoic acid), CNQX (6-cyano-7-nitro-quinoxaline-2,3-dione), and strychnine (to block glutamatergic and glycinergic currents), and 100 nM allopregnanolone (ALLO). After the wash-in of 20 µM bicuculline, the inhibitory postsynaptic currents (IPSCs) suspended and the baseline shifted from −201 pA to −156 pA. (**B**) All-point histogram of the recording presented in (**A**) given as counts per bin, illustrating the shift in the baseline current induced by 20 µM bicuculline in the presence of 100 nM ALLO. The reduction of the standard deviation from 6.1 pA to 4.1 pA is caused by the blockade of synaptic events (IPSCs). (**C**) Summary: ALLO induces tonic currents in spinal interneurons, 50 nM: 9.7 ± 5.3 pA (*n* = 8), *p* = 0.11; 100 nM: 40.8 ± 9.8 pA (*n* = 9); *p* = 0.003 and 250 nM: 49.3 ± 8.2 pA (*n* = 4), *p* = 0.009; horizontal lines = mean, ***** = *p* < 0.05, *t*-test.

**Figure 2 ijms-21-07399-f002:**
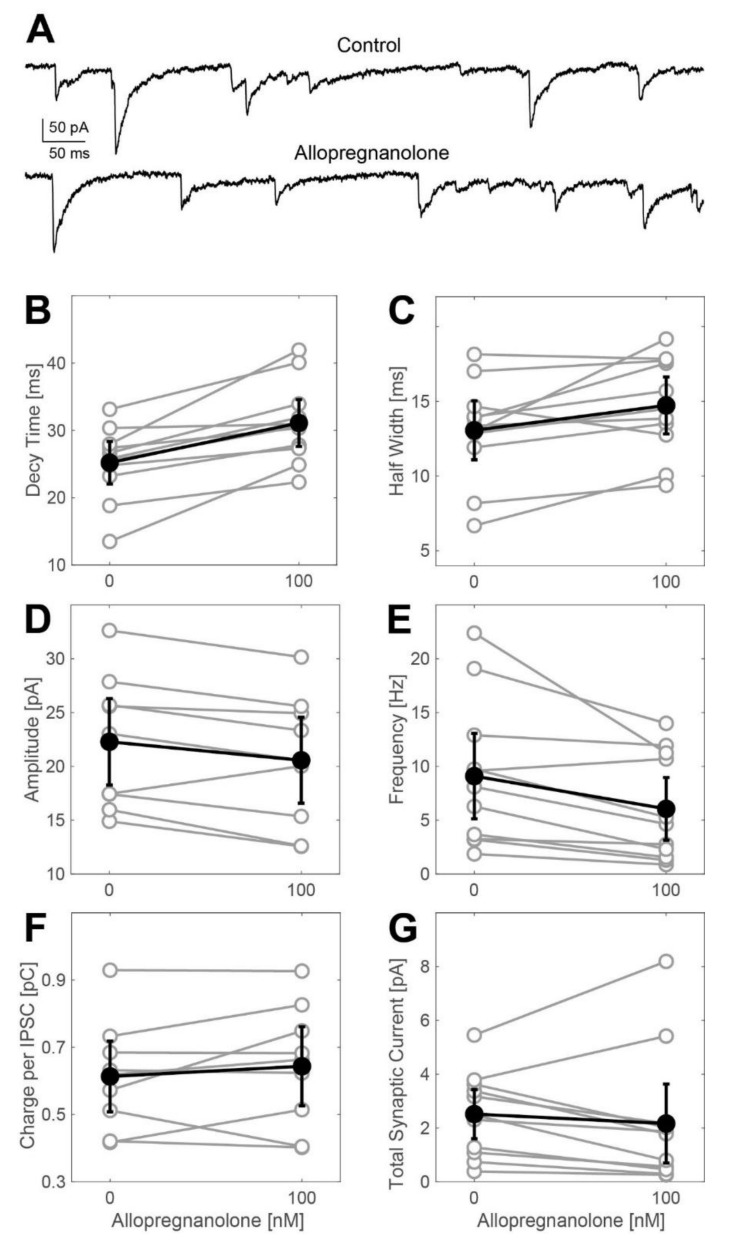
Effects of allopregnanolone (100 nM) on phasic currents in spinal neurons in the ventral horn. (**A**) Original whole-cell voltage-clamp recording from a cultured spinal slice in the absence (upper trace) and presence (lower trace) of ALLO (100 nM). (**B**–**G**) Summary of the actions of ALLO (100 nM) on IPSC decay time (**B**), width (**C**), amplitude (**D**), frequency (**E**), charge per IPSC (**F**), and total synaptic current (**G**). All data are given as the mean including the 95% confidence interval. ALLO at 100 nM induced a significant increase in the IPSC decay time (**B**) from 25.2 (22.03, 28.35) ms to 31.1 (27.6, 34.58) ms (*n* = 11, *p* = 0.0035, *t*-test). The same effect could be detected when analyzing the width of average IPSCs: 100 nM ALLO increases IPSC width (**C**) from 13.1 (11.13, 15.03) ms to 14.7 (12.82, 16.64) ms (*n* = 11, *p* = 0.024, *t*-test). On the other hand, ALLO does slightly reduce the mean amplitude (**D**) of IPSCs: 22.3 (18.27, 26.31) pA at control condition, 20.57 (16.58, 24.56) pA in the presence of 100 nM ALLO (*n* = 9, *p* = 0.02, *t*-test). Furthermore, ALLO significantly reduced IPSC frequency (**E**) from 9.1 (5.13, 13.07) Hz to 6.1 (3.13, 8.97) Hz (*n* = 11, *p* = 0.0005, *t*-test). The net effect on the charge transfer per IPSC (**F**) was approximated by multiplying the decay time and the amplitude of IPSCs, with a result of 0.61 (0.51, 0.72) pC in the absence and 0.64 (0.53, 0.76) pC in the presence of 100 nM ALLO (*n* = 9, *p* = 0.34, *t*-test). (**G**) Summary: when multiplying the charge transfer per IPSC by the IPSC frequency, 100 nM ALLO did not cause any significant increase in the total synaptic current: 2.5 (1.61, 3.44) pC vs. 2.1 (0.63, 3.64) pC (*n* = 9, *p* = 0.42, *t*-test).

**Figure 3 ijms-21-07399-f003:**
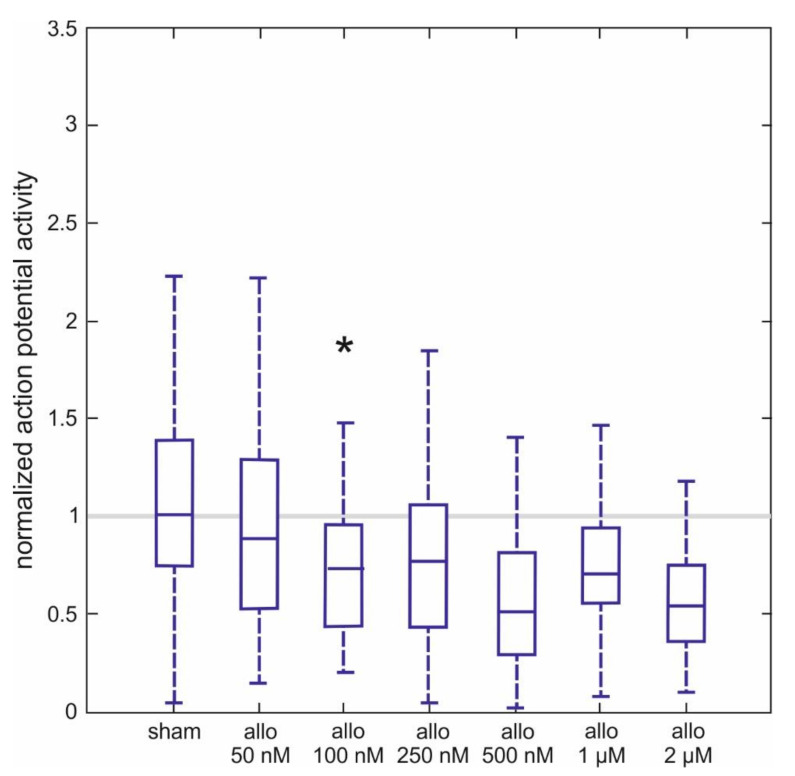
Actions of allopregnanolone on spontaneous action potential frequency of spinal interneurons. Ascending concentrations of ALLO (50 nM–2 µM) induce a concentration-dependent depression of spinal spontaneous network activity (the number of recordings is 35–48 per concentration). As data were not normally distributed (Lilliefors test), they are displayed as boxplots (line: median, box: lower quartile = 25th percentile and upper quartile = 75th percentile; whisker: 1.5*iqr). A significant inhibition was observed, starting from an ALLO concentration of 100 nM (Mann-Whitney U test with Bonferroni correction, ***** = *p* < 0.05).

**Figure 4 ijms-21-07399-f004:**
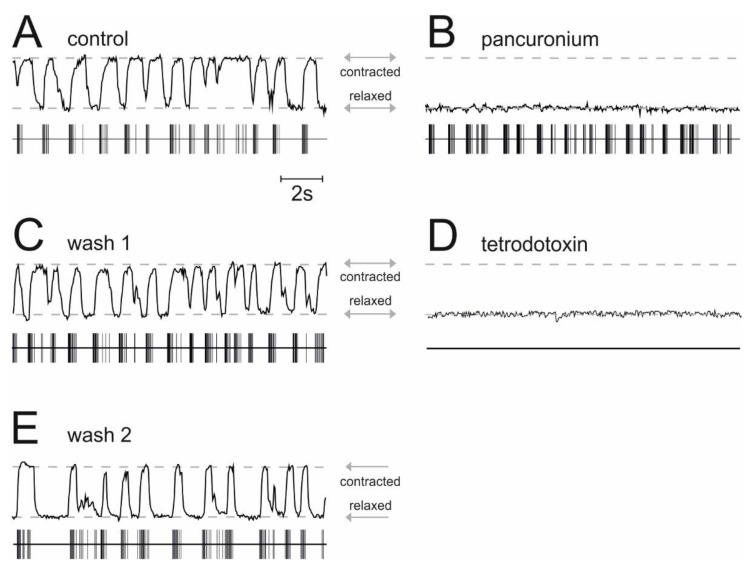
Simultaneous recording of muscle contractions and action potential firing in spinal nerve and muscle co-cultures. The status of the muscle fibers of the nerve and muscle co-culture is given in each figure in the upper half (contracted: line is up, relaxed: line is down), whereas the corresponding neuronal action potential activity is given in the lower half (horizontal line represents baseline noise, each vertical deflection is a single or group of action potentials). (**A**) Control condition in the absence of drugs. Note that there is a correlation between the neuronal activity and the contraction of muscle fibers. (**B**) In the presence of the muscle relaxant pancuronium (1 µM) the muscle activity is down (i.e., relaxed), whereas neuronal activity remains relatively unchanged. (**C**) After the washout of pancuronium, the regained muscle activity is clearly visible. (**D**) Tetrodotoxin (0.5 µM) does not induce any neuronal activity, thus, there is no contraction in the muscle fibers. (**E**) After the washout of tetrodotoxin, neuronal and muscular activity is regained, at least to some extent. Note that there is still a correlation between the neuronal activity and the contraction in the muscle fibers.

**Figure 5 ijms-21-07399-f005:**
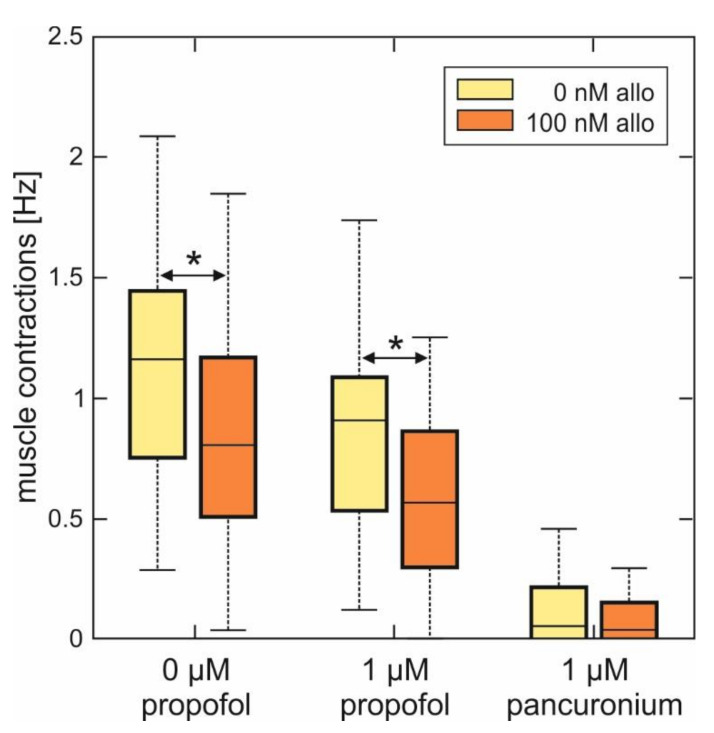
Summary of the effects of allopregnanolone on muscle contraction frequency in spinal nerve and muscle co-cultures in the absence and in the presence of propofol. As data were not normally distributed (Lilliefors test), data are given as boxplots. Left side: 100 nM ALLO significantly reduced the muscle contraction frequency in spinal nerve and muscle co-cultures from 1.16 (0.84, 1.29) Hz under control conditions to 0.81 (0.70, 0.95) Hz (*n* = 29, Mann-Whitney U test, * *p* = 0.043). Middle: while propofol (1 µM) also reduced the median muscle contraction frequency (0.92 (0.55, 1.04) Hz, *n* = 29), this effect significantly increased in the additional presence of 100 nM ALLO: median muscle contraction frequency 0.56 (0.32, 0.78) Hz, *n* = 28, Mann-Whitney U test, * *p* = 0.041, compared with propofol. Right side: muscle contractions are virtually absent in the additional as well as in the sole presence of 1 µM pancuronium.

**Figure 6 ijms-21-07399-f006:**
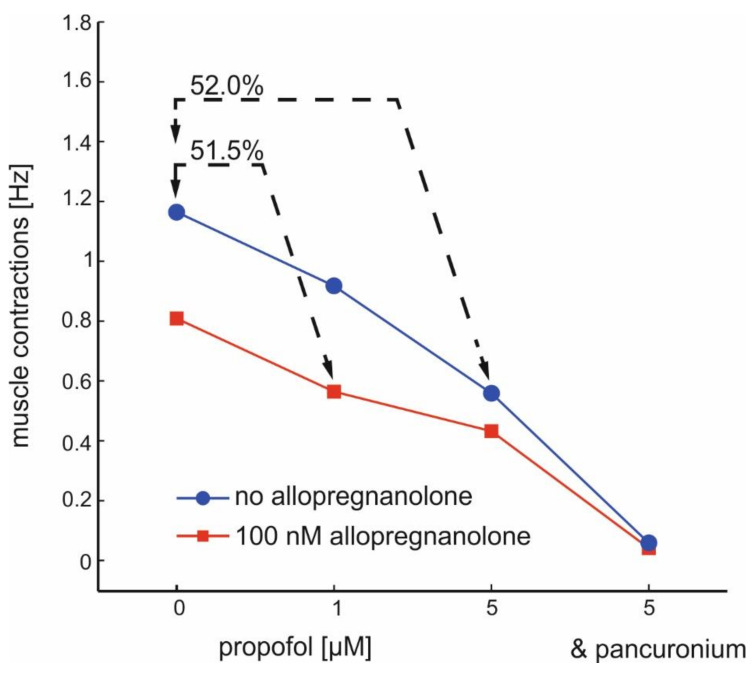
Interactions between allopregnanolone and propofol on the frequency of muscle contractions induced by α-motoneurons. The two lines show the concentration-dependent action of propofol in the absence (blue) and in the presence (red) of ALLO (100 nM). Note that the combined application of 100 nM ALLO and 1 µM propofol reduces the muscle activity to almost the same level as observed after exposure to 5 µM propofol. Furthermore, the efficacy of ALLO in further reducing muscle contractions is low at the highest propofol concentration (5 µM). Muscle contractions are almost completely depressed by pancuronium (1 µM).

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
