# Peer review of "Allopregnanolone Enhances GABAergic Inhibition in Spinal Motor Networks"

_ijms, 2020, doi:10.3390/ijms21197399_

Round 1
Reviewer 1 Report
For this MS, Drexler et al. used advanced electrophysiological techniques to evaluate the interaction between two anesthetics / sedatives in various in vitro slice preparations. In general, the experiments are well performed. This reviewer does not believe that additional experiments are necessary but some additional editorial work would improve the MS. To begin with, the MS would benefit from a succinct underlying hypothesis presented in the introduction (and the abstract), critically tested by the presented experiments and then discussed in the appropriate section.
Introduction
L35 is it known that injectable anesthetics produce immobility by action in the ventral horn? The cited literature does not appear to have that degree of mechanistic resolution.
L59 ref 14 does not test any drug other than propofol, furthermore, tonic current was detectable only at 5 microM which is far above the ‘clinical’ range. Please reconcile.
L62 here and in the whole text, please be more precise in stating what type of neurons (MNs, INs or unidentified) you refer to.
L49-52: general anesthetics include volatile anesthetics. The latter do not ‘dampen nociception in the spinal cord’ (see papers by Jinks). Please provide evidence that injectable anesthetics do exactly that (as opposed to e.g. dampen motor output or brainstem or motorcortex responsiveness).
I do not follow the link between movements intra-op and increased postoperative pain. I am sorry, but I do not see how reference #12 and 13 support the claim made in L51.
Within the limitations imposed by the publisher, I would still prefer to have a brief explanation of organotypic culture properties. In particular, to what degree of certainty motoneurons and different types of interneurons can be identified and to what degree the spinal cord circuitry is preserved in the slice / co-culture. In the opinion of this reviewer, addressing the specifics of this interesting preparations and the characteristics of tonic vs. phasic inhibition would be a better use of the space allotted for the introduction than the current first paragraph which is not really essential for an in-vitro lab-bench study.
Results
L73-78 This section might be better placed in the introduction.
L 114 ‘identical’ to what? Please rephrase.
L 138 – please identify what neurons you refer to
L 168 what is known / has been rigorously documented about connectivity in organotypic slices? You cite publications later on but a statement to that would be hlpful early in the paper.Have MN been identified with any types of specific (immunocytochemistry, gene expression etc) markers? Has your lab ever confirmed the findings originally described in the 90ies?
L 180. The effect of pancuronium actually only proves that it's via nicotinic receptors with the additional caveat that pancuronium has also sympathomimetic effects meaning that it is not a highly selective (note that other neuromuscular blocker do not have this effect so it’s unlikely to be mediated by nAChR blockade at the NMJ). You would have to close the loop of argumentation and prove / argue that the only source for ACh in the spinal cord are MNs. Furthermore, why not use a more specific nAChR blocker than pancuronium?
L 206: % and fraction is used unsystematically. Please settle on one or the other (or both) but use consistently
L207 The effect of pancuronium on muscle contraction is actually unrelated to the topic of the paper.
Discussion
L245-248 would be useful in the introduction to set up the framework for the experiments and could be explained in more detail here mentioning how the similarity of the circuitry and the identity of relevant neuronal populations has been proven.
L339 This is a far reaching (and at least partially inaccurate) statement to be supported by the quoted references. Ref 33 is a ‘negative’ while #34 only addresses propofol.
A potential general limitation of the experimental approach which does not invalidate the findings but should be considered if extrapolations to the in-vivo situation are made is that recordings of synaptic activity under conditions of blockade of all excitatory synaptic activity (which seems to be the case in all the reported experiments) are an imperfect image of reality at best. The interruption of physiological circuitry probably has profound implications for the balance between phasic and tonic activity in their overall effect on neuronal excitability, distorting the contribution of the two components.
For this reason and because of the general awkwardness of using in vitro findings to justify conclusions for in vivo drug use, I would recommend abstaining from any speculations about clinical utility (L339-end). Instead, proposing experiments that could actually test the hypothesis that modulation of tonic GABAergic activity (when performed in vivo) would actually influence excitability of any elements in the spinal cord circuitry thereby contributing to immobility under anesthesia would be a welcome addition to the discussion.
Author Response
Dear Patricia Csegezi,
Dear Prof. Rammes,
Dear Editor,
Thank you for sending us the reviewers’ comments on our manuscript ijms-931644 entitled "Allopregnanolone predominately depresses spinal network activity by
tonic GABAergic inhibition thereby fostering propofol-induced neuro-muscular
relaxation".
We thank the reviewers for providing valuable advice. Here we present a revised version of our manuscript, which addresses the weaknesses pointed out by the reviewers. All changes in the manuscript are highlighted by blue color.
We would like to answer the reviewer’s comments as follows:
Reviewer #1:
For this MS, Drexler et al. used advanced electrophysiological techniques to evaluate the interaction between two anesthetics / sedatives in various in vitro slice preparations. In general, the experiments are well performed. This reviewer does not believe that additional experiments are necessary but some additional editorial work would improve the MS. To begin with, the MS would benefit from a succinct underlying hypothesis presented in the introduction (and the abstract), critically tested by the presented experiments and then discussed in the appropriate section.
In the new version of our manuscript we now present a working hypothesis in the introduction (see page 2, line 46 et seqq.): Thus, our study is testing the hypothesis that ALLO attenuates action potential firing of spinal interneurons and motoneurons predominantly via enhancing tonic but not synaptic GABAergic inhibition. Furthermore, we are addressing the question whether ALLO and propofol, the most frequently used intravenous anesthetic, affect spinal motoneurons in a synergistic manner as previously reported for synaptically mediated GABAergic currents in neocortical neurons and a1b2g2L GABAA receptors activated by exposure to low GABA-concentrations.
The central working hypothesis of our current study is also given in the abstract, please see page 1, line 8 et seqq.
The results are discussed in section 3.1: Allopregnanolone induces a tonic GABAA-R mediated current and also in section 3.2. Actions of allopregnanolone on muscle contractions. The new section 3.5. deals with the limitations of our experimental approach.
Introduction
L35 is it known that injectable anesthetics produce immobility by action in the ventral horn? The cited literature does not appear to have that degree of mechanistic resolution.
We agree. The previously cited papers only provide data on the incidence of intraoperative movements. In the revised version of our manuscript we removed the whole paragraph according to reviewer#2’s request to focus on allopregnanolone and to drop content that is related to propofol.
L59 ref 14 does not test any drug other than propofol, furthermore, tonic current was detectable only at 5 microM which is far above the ‘clinical’ range. Please reconcile.
The wording was indeed mistakable, as we silently assume that propofol, midazolam and etomidate act on the molecular level via targets that are different from neurosteroid-sensitive receptors. According to the proposal of reviewer#1 not to speculate about potential clinical applications, this paragraph was omitted.
L62 here and in the whole text, please be more precise in stating what type of neurons (MNs, INs or unidentified) you refer to.
All electrophysiological recordings were carried out on ventral horn interneurons. Motoneuron-activity was assessed indirectly via video-recording of muscle contractions. This is now clearly stated throughout the revised manuscript.
L49-52: general anesthetics include volatile anesthetics. The latter do not ‘dampen nociception in the spinal cord’ (see papers by Jinks). Please provide evidence that injectable anesthetics do exactly that (as opposed to e.g. dampen motor output or brainstem or motorcortex responsiveness).
Again, the reviewer is correct. Nociception should be replaced by painful stimuli-induced motor reflexes. Steven L. Jinks’ paper nicely shows that locomotor circuits in the ventral spinal cord are the primary site of action for isoflurane and halothane to ablate motor reflexes. This also seems to apply to propofol [1]. In the revised version of our manuscript, this part in the introduction section was replaced by a detailed description of our preparation, as recommended by this reviewer.
I do not follow the link between movements intra-op and increased postoperative pain. I am sorry, but I do not see how reference #12 and 13 support the claim made in L51.
The simple idea is that the occurrence of intraoperative painful-stimuli induced movements is indicative for insufficient depression of nociception in the dorsal horn, which should be achieved by opioids. The lack of intraoperative analgesia in turn causes increased postoperative pain, which does not seem to be totally unlikely. References #12 and #13 only provided evidence that postoperative pain is indeed a problem in clinical anesthesia. However, we removed these speculations about potential clinical use from the revised version of our manuscript, as recommended by this reviewer.
Within the limitations imposed by the publisher, I would still prefer to have a brief explanation of organotypic culture properties. In particular, to what degree of certainty motoneurons and different types of interneurons can be identified and to what degree the spinal cord circuitry is preserved in the slice / co-culture. In the opinion of this reviewer, addressing the specifics of this interesting preparations and the characteristics of tonic vs. phasic inhibition would be a better use of the space allotted for the introduction than the current first paragraph which is not really essential for an in-vitro lab-bench study.
In the revised version of our manuscript, we have included a more detailed description of our organotypic cultures in the introduction, see page 2, line 51 et seqq.
Results
L73-78 This section might be better placed in the introduction.
Done.
L 114 ‘identical’ to what? Please rephrase.
This should read: “…were almost identical to those observed at a concentration of 100 nM.” (see page 5, line 118 in the revised version of our manuscript.)
L 138 – please identify what neurons you refer to
These recordings were carried out on interneurons in the ventral part of spinal tissue cultures. This information was added in the revised version of our manuscript, see page 7, line 143.
L 168 what is known / has been rigorously documented about connectivity in organotypic slices? You cite publications later on but a statement to that would be helpful early in the paper. Have MN been identified with any types of specific (immunocytochemistry, gene expression etc.) markers? Has your lab ever confirmed the findings originally described in the 90ies?
Motoneurons had been labeled by retrograde axonal staining. Small horseradish peroxidase crystals were topically applied on he muscle tissue. Using this approach, the authors labelled large, multipolar cells located at the ventral border of the cultured spinal tissue [2]. Furthermore, motoneurons were labelled by acetylcholinesterase-silver staining [2]. In addition, Streit and coworkers [3] performed intracellular recordings and characterized the membrane and firing properties of these putative motoneurons, which can be easily discriminated from other spinal neurons by their size and location. About 10 years later Avossa and coworkers investigated spatial and temporal regulation of neuronal and non-neuronal markers by immunocytochemical and Western blotting analysis using antibodies against neurofilament H, choline acetyltransferase, glutamic acid decarboxylase 67, HERG, glial fibrillary acidic protein and myelin basic protein [4]. Many years ago we confirmed retrograde labelling of motoneurons in our lab. But it is a good idea to repeat this. We have now included this information in the introduction and also in the discussion section.
L 180. The effect of pancuronium actually only proves that it's via nicotinic receptors with the additional caveat that pancuronium has also sympathomimetic effects meaning that it is not a highly selective (note that other neuromuscular blocker do not have this effect so it’s unlikely to be mediated by nAChR blockade at the NMJ). You would have to close the loop of argumentation and prove / argue that the only source for ACh in the spinal cord are MNs. Furthermore, why not use a more specific nAChR blocker than pancuronium?
In the present study, pancuronium was applied at the very end of every individual experiment in order to confirm that recorded muscle contractions were neurogenic in origin. This was the only purpose of using pancuronium. In a minority of experiments (about 20% of all tested preparations), muscle contractions persisted even after exposure to pancuronium. In these cases, spontaneous muscle contractions most probably resulted from a depolarized membrane potential of muscle fibers but not motoneuronal synaptic input. When these pancuronium-resistant fibers were exposed to the depolarizing muscle relaxant succinylcholine, contractions immediately stopped (data not shown), suggesting that in the absence of cholinergic input, a nicotinic agonist was still capable to produce depolarization-induced muscle relaxation. However, pancuronium-resistant muscle activity does not appear to reflect motoneuronal activity. Therefore, these experiments were excluded from further analysis. When pancuronium was applied to co-cultures of spinal cord and muscle tissue, action potential firing of spinal interneurons was not significantly reduced, suggesting that the drug had only minor effects on spinal neurons (see Figure 4B). We used pancuronium because of the pre-existing literature when starting this project many years ago. Using the phrenic nerve diaphragm preparation, Seeger T et al 2007 showed that at a concentration of 1 µM (the same concentration also used in our study) pancuronium completely depressed muscle contractions evoked by electric stimulation of efferent cholinergic fibers [5]. When the authors substantially increased stimulus strength, resulting into direct depolarization of the membrane potential of muscle fibers, contractions re-appeared, suggesting that pancuronium only depressed synaptic transmission at the neuromuscular endplate, but had no direct inhibitory effect on the membranes of muscle fibers. We appreciate the reviewer’s remark on pancuronium’s well-known side effects, which are probably mediated by muscarinic receptors [6]. However, administration of the muscarinic blocker atropine in the high concentration range never abolished muscle contractions (we are planning to present these data in a following paper).
Taken together, the observation that in our preparation muscle fibers are only innervated by acetylcholinesterase-positive, large neurons located at the ventral border of cultured spinal slices [2], in combination with the use of only pancuronium-sensitive muscle contractions for further statistical analysis, implicates that this muscle activity is closely linked to the activity of spinal motoneurons. We do not believe that this conclusion needs to be further supported by additional experiments, but we welcome the reviewer’s suggestion to switch to a less dirty non-depolarizing muscle relaxant in the near future.
L 206: % and fraction is used unsystematically. Please settle on one or the other (or both) but use consistently
We do not use fraction in our manuscript. We use absolute frequency of muscle contractions, given in Hertz [Hz] and relative decrease of muscle concentration to illustrate the effect of a drug, given in percent [%]. As the frequency of muscle contractions under control condition is somehow around 1 Hz, the values given might be regarded as normalized values by mistake. We added the information “[Hz]” to every single value of muscle fiber contraction frequency to avoid such misunderstanding in the revised version of our manuscript.
L207 The effect of pancuronium on muscle contraction is actually unrelated to the topic of the paper.
We do not agree. As outlined in detail above, the use of pancuronium is in our opinion important to confirm that we actually did monitor neurogenic-induced muscle contractions.
Discussion
L245-248 would be useful in the introduction to set up the framework for the experiments and could be explained in more detail here mentioning how the similarity of the circuitry and the identity of relevant neuronal populations has been proven.
Done. (Please see above.)
L339 This is a far reaching (and at least partially inaccurate) statement to be supported by the quoted references. Ref 33 is a ‘negative’ while #34 only addresses propofol.
This sentence has been removed.
A potential general limitation of the experimental approach which does not invalidate the findings but should be considered if extrapolations to the in-vivo situation are made is that recordings of synaptic activity under conditions of blockade of all excitatory synaptic activity (which seems to be the case in all the reported experiments) are an imperfect image of reality at best. The interruption of physiological circuitry probably has profound implications for the balance between phasic and tonic activity in their overall effect on neuronal excitability, distorting the contribution of the two components.
In the revised version of our manuscript this limitation is discussed in the newly added chapter 3.5. Limitations of the present study.
For this reason and because of the general awkwardness of using in vitro findings to justify conclusions for in vivo drug use, I would recommend abstaining from any speculations about clinical utility (L339-end). Instead, proposing experiments that could actually test the hypothesis that modulation of tonic GABAergic activity (when performed in vivo) would actually influence excitability of any elements in the spinal cord circuitry thereby contributing to immobility under anesthesia would be a welcome addition to the discussion.
At this point we somehow disagree with the reviewer’s statement that there is a ‘general awkwardness of using in vitro findings’ to make conclusions for in vivo drug use. A better understanding of the molecular, cellular and network mechanism by which drugs act provides novel perspectives in the interpretation of clinical drug actions even for experienced physicians. Furthermore, the relevance of in vitro findings is supported by the increased use of complex in vitro systems (as utilized here) in drug development and by the widely accepted recommendation to replace animal experimentation by appropriate in vitro methods [7]. More specifically for our preparation, considerable work has been carried out in the past decades in order to compare the properties of this in vitro preparation to the in vivo properties of spinal neurons and muscle fibers, with respect to the morphology of cell types, use of neurotransmitters, synaptic connectivity, developmental markers and sensitivity of drugs. All these studies implicate that spinal organotypic cultures are useful in vitro models. As an example, we reported that the intravenous anesthetics etomidate is, at clinically relevant concentrations, effective in spinal cord cultures derived from wild type but not in tissue slices derived from (N265M)b3-knockin mice [8]. In vitro actions on neuronal discharge rate and GABAergic transmission correlated well with the drugs’ ability in depressing painful-stimuli induced movements in wild type animals but not b3-knockin mice, confirming a high predictive power of our in vitro results. However, with mixed feelings we follow reviewer’s request to remove all our suggestions for in vivo drug use in this manuscript for two reasons. First, this is to our best knowledge the very first study evaluating the effect of a neurosteroid on spinal motor circuits. Thus, confirming or falsifying implications arising from the present work can be easily done by further investigating the actions of additional neurosteroids and TSPO-agonists. Indeed, this is an important issue to do. Likewise, confirming implications drawn from the present study for in vivo drug use by specifically designed animal experimentation is also expected to increase our finding’s explanatory power. Therefore, we also followed the reviewer’s advice to propose future experiments in the discussion section.
Best regards,
Berthold Drexler and Bernd Antkowiak
on behalf of the authors
- Kungys, G.; Kim, J.; Jinks, S. L.; Atherley, R. J.; Antognini, J. F., Propofol produces immobility via action in the ventral horn of the spinal cord by a GABAergic mechanism. Anesth. Analg 2009, 108, (5), 1531-1537.
- Spenger, C.; Braschler, U. F.; Streit, J.; Lüscher, H.-R., An organotypic spinal cord - dorsal root ganglion - skeletal muscle coculture of embryonic rat. I. The morphological correlates of the spinal reflex arc. European Journal of Neuroscience 1991, 3, 1037-1053.
- Streit, J.; Spenger, C.; Lüscher, H.-R., An organotypic spinal cord - dorsal root ganglion - skeletal muscle coculture of embryonic rat. II. Functional evidence for the formation of spinal reflex arcs in vitro. European Journal of Neuroscience 1991, 3, 1054-1068.
- Avossa, D.; Rosato-Siri, M. D.; Mazzarol, F.; Ballerini, L., Spinal circuits formation: a study of developmentally regulated markers in organotypic cultures of embryonic mouse spinal cord. Neuroscience 2003, 122, (2), 391-405.
- Seeger, T.; Worek, F.; Szinicz, L.; Thiermann, H., Reevaluation of indirect field stimulation technique to demonstrate oxime effectiveness in OP-poisoning in muscles in vitro. Toxicology 2007, 233, (1-3), 209-213.
- Cembala, T. M.; Sherwin, J. D.; Tidmarsh, M. D.; Appadu, B. L.; Lambert, D. G., Interaction of neuromuscular blocking drugs with recombinant human m1-m5 muscarinic receptors expressed in Chinese hamster ovary cells. Br J Pharmacol 1998, 125, (5), 1088-94.
- Beilmann, M.; Boonen, H.; Czich, A.; Dear, G.; Hewitt, P.; Mow, T.; Newham, P.; Oinonen, T.; Pognan, F.; Roth, A.; Valentin, J. P.; Van Goethem, F.; Weaver, R. J.; Birk, B.; Boyer, S.; Caloni, F.; Chen, A. E.; Corvi, R.; Cronin, M. T. D.; Daneshian, M.; Ewart, L. C.; Fitzgerald, R. E.; Hamilton, G. A.; Hartung, T.; Kangas, J. D.; Kramer, N. I.; Leist, M.; Marx, U.; Polak, S.; Rovida, C.; Testai, E.; Van der Water, B.; Vulto, P.; Steger-Hartmann, T., Optimizing drug discovery by Investigative Toxicology: Current and future trends. ALTEX 2019, 36, (2), 289-313.
- Grasshoff, C.; Jurd, R.; Rudolph, U.; Antkowiak, B., Modulation of presynaptic beta3-containing GABAA receptors limits the immobilizing actions of GABAergic anesthetics. Mol. Pharmacol 2007, 72, (3), 780-787.
Reviewer 2 Report
This study is difficult to read I agree this is a very specialized subject but at least rationales should be clearly explained
the main concern is why this finding should be coupled with propofol effect?
If the finding is new (the authors claim) therefore this information (reduction action potential firing of spinal interneurons) is sufficient to be published and there was no need to be coupled with propofol
Introduction:
The rationale and primary and secondary objective should be clearly stated.
Results:
figure 1 c please explain the asterix
Discussion
material and methods
If possible images or video of the visualization of muscle activity should be available perhaps with a link or appendice
The quality control algorithm should also be exposed permitting other researchers to perform such study
Author Response
Dear Patricia Csegezi,
Dear Prof. Rammes,
Dear Editor,
Thank you for sending us the reviewers’ comments on our manuscript ijms-931644 entitled "Allopregnanolone predominately depresses spinal network activity by
tonic GABAergic inhibition thereby fostering propofol-induced neuro-muscular
relaxation".
We thank the reviewers for providing valuable advice. Here we present a revised version of our manuscript, which addresses the weaknesses pointed out by the reviewers. All changes in the manuscript are highlighted by blue color.
We would like to answer the reviewer’s comments as follows:
Reviewer #2:
This study is difficult to read I agree this is a very specialized subject but at least rationales should be clearly explained
In the revised version of the manuscript the study’s rationales are now clearly stated in the introduction.
the main concern is why this finding should be coupled with propofol effect?
If the finding is new (the authors claim) therefore this information (reduction action potential firing of spinal interneurons) is sufficient to be published and there was no need to be coupled with propofol
After re-reading the manuscript we agree that it wasn’t a good idea referring so often to propofol’s effects, in particular because most of propofol’s actions have been described in a previous paper. Accordingly, we changed the title and alter the text by removing most of the respective paragraphs. However, since ref#1 pointed out that this is a lab bench study and its relation to the effects of allopregnanolone in vivo remains to be elucidated in future studies, we would like to present the effects of the gold-standard-intravenous anesthetic propofol on the frequency of muscle contractions in order to enable the comparison between the effects of allopregnanolone and a clinically used anesthetic.
Introduction:
The rationale and primary and secondary objective should be clearly stated.
Done, please see above.
Results:
figure 1 c please explain the asterix
Done. * = p < 0.05, t-test.
Discussion
material and methods
If possible images or video of the visualization of muscle activity should be available perhaps with a link or appendice.
This is just a very good idea. A video can be found in the additional material / supplements uploaded with this manuscript (CoCultures_Final2.mp4).
The quality control algorithm should also be exposed permitting other researchers to perform such study.
We thank the reviewer for this suggestion. We describe our quality control algorithm in great detail in the revised version of our manuscript in the Material and Methods section, paragraph 4.4., page 15, line 450 et seqq., starting with "For reducing data variability..."
Best regards,
Berthold Drexler and Bernd Antkowiak
on behalf of the authors
Literature
- Kungys, G.; Kim, J.; Jinks, S. L.; Atherley, R. J.; Antognini, J. F., Propofol produces immobility via action in the ventral horn of the spinal cord by a GABAergic mechanism. Anesth. Analg 2009, 108, (5), 1531-1537.
- Spenger, C.; Braschler, U. F.; Streit, J.; Lüscher, H.-R., An organotypic spinal cord - dorsal root ganglion - skeletal muscle coculture of embryonic rat. I. The morphological correlates of the spinal reflex arc. European Journal of Neuroscience 1991, 3, 1037-1053.
- Streit, J.; Spenger, C.; Lüscher, H.-R., An organotypic spinal cord - dorsal root ganglion - skeletal muscle coculture of embryonic rat. II. Functional evidence for the formation of spinal reflex arcs in vitro. European Journal of Neuroscience 1991, 3, 1054-1068.
- Avossa, D.; Rosato-Siri, M. D.; Mazzarol, F.; Ballerini, L., Spinal circuits formation: a study of developmentally regulated markers in organotypic cultures of embryonic mouse spinal cord. Neuroscience 2003, 122, (2), 391-405.
- Seeger, T.; Worek, F.; Szinicz, L.; Thiermann, H., Reevaluation of indirect field stimulation technique to demonstrate oxime effectiveness in OP-poisoning in muscles in vitro. Toxicology 2007, 233, (1-3), 209-213.
- Cembala, T. M.; Sherwin, J. D.; Tidmarsh, M. D.; Appadu, B. L.; Lambert, D. G., Interaction of neuromuscular blocking drugs with recombinant human m1-m5 muscarinic receptors expressed in Chinese hamster ovary cells. Br J Pharmacol 1998, 125, (5), 1088-94.
- Beilmann, M.; Boonen, H.; Czich, A.; Dear, G.; Hewitt, P.; Mow, T.; Newham, P.; Oinonen, T.; Pognan, F.; Roth, A.; Valentin, J. P.; Van Goethem, F.; Weaver, R. J.; Birk, B.; Boyer, S.; Caloni, F.; Chen, A. E.; Corvi, R.; Cronin, M. T. D.; Daneshian, M.; Ewart, L. C.; Fitzgerald, R. E.; Hamilton, G. A.; Hartung, T.; Kangas, J. D.; Kramer, N. I.; Leist, M.; Marx, U.; Polak, S.; Rovida, C.; Testai, E.; Van der Water, B.; Vulto, P.; Steger-Hartmann, T., Optimizing drug discovery by Investigative Toxicology: Current and future trends. ALTEX 2019, 36, (2), 289-313.
- Grasshoff, C.; Jurd, R.; Rudolph, U.; Antkowiak, B., Modulation of presynaptic beta3-containing GABAA receptors limits the immobilizing actions of GABAergic anesthetics. Mol. Pharmacol 2007, 72, (3), 780-787.
Round 2
Reviewer 2 Report
I thank the authors
They substantially improved the manuscript , which is now much easier to read while they answered adequately to most of queries